# Effect of HHP, Enzymes and Gelatin on Physicochemical Factors of Gels Made by Using Protein Isolated from Common Cricket (*Acheta domesticus*)

**DOI:** 10.3390/foods10040858

**Published:** 2021-04-15

**Authors:** Pietro Urbina, Cuauhtemoc Marin, Teresa Sanz, Dolores Rodrigo, Antonio Martinez

**Affiliations:** Instituto de Agroquimica y Tecnología de Alimentos CSIC, Paterna, 46980 Valencia, Spain; jpietroup@gmail.com (P.U.); neodave@gmail.com (C.M.); tesanz@iata.csic.es (T.S.); lolesra@iata.csic.es (D.R.)

**Keywords:** high hydrostatic pressure, gelation, enzymes, cricket protein, gelatin, high protein food

## Abstract

The effect of high hydrostatic pressure (HHP) combined with enzymatic methods or gelatin incorporation in the gelation process of protein isolated from Acheta domesticus was investigated. The results indicate that transglutaminase (TGasa) or glucose oxidase (GOx) enzymes can induce reversible aggregation in dispersions of insoluble protein fractions and increase viscosity in dispersions of soluble fractions, but does not induce gel formation even after HHP treatment; in consequence, enzymatic treatment on cricket protein can be used to increase viscosity but not to form gels. It is technically feasible to obtain gels by adding 2% porcine gelatin to dispersions of protein fractions and subjecting them to HHP. The firmness and syneresis variation values of those gels during storage depended on the protein extracted fraction (insoluble or soluble protein) and on the concentration of protein used. The highest hardness and lowest syneresis was found with the gels obtained from the insoluble fraction at 11 and 15% (*w*/*w*) protein concentration. Color difference (*ΔE** > 3) appreciable to the naked eye was observed along the storage period and no noticeable pH variations were found after 28 days of storage. Results indicate that new raw materials of interest can be developed for the food industry based on cricket protein isolates, to make high-protein foods which could be applied in a wide variety of different food applications including 3D printing or fat substitution.

## 1. Introduction

High Hydrostatic Pressure is a novel preservation technology that has experienced a great development in recent years. However, this technology has other applications in the food sector. It has been employed for protein gelation purposes as an alternative to heat processing in denaturing proteins, and gel formation. Soy protein formed gels when high hydrostatic pressure was applied with an intensity of 300 MPa [1]. A study to elucidate the effect of high-pressure treatment (200, 400, and 600 MPa) on the physicochemical, functional, thermal, rheological, and structural properties of freeze-dried kidney bean protein isolate was carried out by [2]. The effects of different non thermal technologies on the physicochemical and functional properties of myofibrillar proteins were studied by [3]. The results indicated that high pressure processing was the most effective technique in modifying myofibrillar proteins improving the yield of gels when heated. Nevertheless, there are no works in the scientific literature where the use of high hydrostatic pressure combined with other physical or chemical agents on the formation of gels of isolated insect protein was explored.

Edible insects such as *Acheta domesticus* (Common cricket) or *Tenebrio molitor* (Yellow mealworm) are a rich source of protein and yield higher protein content compared to various animals’ meat and consuming a smaller amount of feed, water and land, producing lower environmental impacts. In most countries where edible insects form part of the diet, they are consumed whole after being cooked. Besides consuming edible insects (cooked, fried, or grilled), insect-based products are currently being developed in more complex and processed matrices [4]. As a proof of the interest for insects as food, EFSA [5] adopted an opinion in relation to the safety of dried yellow mealworm as a novel food. Nevertheless, consumers in developed countries, in general, reject the idea of consuming edible insects as such and, therefore, the development of new foods based on isolated insect protein or derived flour presents a new business opportunity. Such products should be better accepted by consumers, overcoming existing consumer reticence.

Insect flour has been incorporated in different food products and some research has been carried out regarding the use of cricket (*Acheta domesticus*) protein foodstuffs. A partial replacement of corn flour by the common cricket (*Acheta domesticus)* in tortillas (a staple of the Central American diet) was carried out by [6]. The results indicated that at 20% replacement by the cricket flour, the color and texture properties as well as the cooking time were optimal, and the protein level was enriched. A granola bar based on flour from *A. domesticus* as an alternative to improve protein supplementation in Colombia was formulated by [7].

The effect of incorporating cricket flour in extruded rice products has also been studied, observing that with substitutions of 10 to 15% the final product was softer, darker, more adhesive with higher protein levels than the original product; furthermore its sensory acceptability was also similar to that of the original product [8]. Substituting 10% of the meat content with *A. domesticus* flour, the mineral and protein content of the meat emulsion increased while the techno-functional properties of the flour remained almost unchanged [9].

Such developments have sparked an interest in the evaluation of the techno-functional properties of isolated insect proteins as a way to find new functional properties which would take place in new food applications [10]. Foam formation and emulsion stabilization properties of cricket protein subjected to partial hydrolysis was studied by [11]. Results indicated that mild hydrolysis improved these properties but could affect stability during storage. A study comparing the solubility properties at different pH levels, oil and water retention; emulsifying activity and foam formation and stabilization of proteins from three insect species including *G. Sigillatus* was carried out by [12]. The authors concluded that proteins were soluble in a broader pH range and that, in general, the proteins evaluated from various species had good techno-functional properties compared to other proteins used in food production, with the exception of foaming. From the aforementioned studies it appears clear that edible insect proteins has the potential to be incorporated into foods with a good supply of proteins and other nutrients, adequate techno-functional characteristics and in products which are part of the consumer’s usual diet. These kinds of studies are opening up the market for cricket-based products, paving the way for increased commercial offer and product diversification [13].

In the present work, the effect of high hydrostatic pressure combined with enzymes and gelatin to promote gel formation by using protein isolated from *Acheta domesticus* was studied. Gel stability and variation of some physicochemical properties during storage was investigated. This research is a first attempt towards the employment of insect proteins in food gels applications as novel foods.

## 2. Materials and Methods

### 2.1. Protein Extraction

The first step in this study was to isolate protein from the common cricket, *A. domesticus.* Cricket flour (Thailand Unique Brand, Thailand) was used for this purpose. Two methods of extraction were carried out; the acid extraction method described by [14] using the same pH level and centrifugation parameters used by those authors. The alkaline extraction was similar to the acid extraction procedure, but the parameters, pH, centrifugation speed, time and temperature were modified according to preliminary tests enabling the optimization of the extraction performance to achieve the highest protein content in the pellet.

For each extraction type (acid or alkaline) 400 g of cricket flour (Thailand Unique Brand) were dispersed in 1200 mL of distilled water. Subsequently the pH was adjusted by adding ascorbic acid or NaOH (Scharlab Chemie S. A., Barcelona, Spain) according to the extraction procedure, acid pH 5 or alkaline pH 10, respectively. The resulting dispersion was stirred on a magnetic stirrer for one hour to achieve a good dispersion and then filtered through a 400 µm light metal sieve. The residue remaining in the sieve was distributed in falcon tubes and frozen at −80 °C for posterior liophylization. The filtrate was centrifuged in a Beckman J-5 centrifuge applying the following conditions: acid extraction a pH of 5, 15,000 g, at 4 °C for 30 min and for alkaline extraction a pH of 10, 4200 g, at 4 °C for 10 min. In both cases, two fractions, an insoluble pellet and a soluble supernatant were obtained. For the two extractions, the fatty matter that remains floating in the supernatant is removed with a spoon. The two fractions of each extraction were frozen at −80 °C and lyophilized at the same time as the previously stored frozen residue.

### 2.2. Quantification of the Percentage of Proteins in Each Fraction

The test was carried out in triplicate for each sample (residual, pellet and supernatant of each extraction) using a nitrogen analyzer kit based on the Dumas method (Elementar, Langenselbold, Germany). To do so, 500 mg of sample were placed inside a tin capsule and subjected to combustion. The gases produced were purified by a catalysts obtaining N2, among other gases. This was passed through a chromatography column and quantified using a thermal conductivity detector (TCD). The calibration curve for the determination was made with an aspartic acid standard. Total nitrogen results were multiplied by factor 6.25 to estimate protein content.

### 2.3. Gelation Study of Cricket Protein Fractions Treated with Enzymes, Heat and High Hydrostatic Pressure (HHP)

The protein fractions (residue, pellet, supernatant), extracted by acid and alkaline methods, were used to prepare two batteries of test tubes of 10 mL each as follows: Protein fraction 1.5 g (15% *w*/*w*), NaCl 0.25 g (2.5% *w*/*w*), CaCl_2_ 0.25 g (2.5% *w*/*w*), sodium-phosphate (Na_2_HPO4/NaH_2_PO_4_) buffer 8.5 mL and 10 mg (0.1% *w*/*w*) of enzyme transglutaminase (TGase) to catalyse acyl group transfer reactions and induction of covalent cross-links in proteins or glucose oxidase (GOx) so that in protein systems, the peroxide resulting from the treatment with GOx, acts as an inducer of disulphide bond formation (Scharlab Chemie S. A., Barcelona, Spain).

Whey protein isolate (WPI) was used as reference of gelation for comparative purposes. In this case, the buffer applied as diluent was sodium-phosphate (Na_2_HPO4/NaH_2_PO_4_) (Scharlab Chemie S. A., Barcelona, Spain) at 0.1 M concentration. All samples were incubated at 50 °C for 16 h. At this temperature, both enzymes have good activity as studied by [15,16]. Then, they were kept under refrigeration at 4 °C to stop enzymatic activity and to avoid alteration until be tested. After this storage period, samples were subjected to a water-bath heat process at 85 °C for 15 min to allow gelation. Non gelled samples were treated by high hydrostatic pressure (HHP) to promote gelation. The HHP treatment conditions were 500 MPa for 15 min at 25 °C, based on gelation studies by [17] on isolated whey protein.

### 2.4. Gelation Study of Cricket Protein Fractions by the Addition of Porcine Gelatin (E441) and Treatment by High Hydrostatic Pressure (HHP)

Gels were formulated with the different cricket protein fractions (pellet and supernatant) obtained by acid and alkaline extraction methods. The same protein isolated and salts levels of each fraction used by [17] were selected: 7, 11 and 15% (*w*/*w*). For that, 2.8, 4.4 and 6 g of isolated cricket protein were added to 40 mL of solution consisting of: NaCl 0.1 g (0.25% *w*/*w*), CaCl_2_ 0.06 g (0.15% *w*/*w*), porcine gelatin E441 0.8 g (2% *w*/*w*). The porcine gelatin E441 dose was selected based on previous experiments considering the minimum level at which a firm gel could be obtained. The resulting solutions containing the protein were dispensed into eppendorf tubes and vacuum sealed in vacuum special coextruded ROVAC ^®^ films bags, kept refrigerated at 4 °C for 16 h, and then HHP treated at 500 MPa for 15 min at 25 °C [17]. After treatment, the samples were again stored in refrigeration. A non HHP-treated sample was considered for comparative purposes.

### 2.5. Physical Characterization of Samples

#### 2.5.1. pH Measurement

Suspensions of 1.5 g of gel in 6 mL of H_2_O were prepared according to the method of [18]. The suspensions were homogenized in a vortex and the pH was measured in triplicate with a potentiometer (Crison Basic 20+, Barcelona, Spain).

#### 2.5.2. Color Measurement

It was determined using a Konica Minolta colorimeter (Konica Minolta Sensing Inc., Tokyo, Japan). Five measurements were taken for each sample. The coordinate results of the Hunter color system (a*, b*, and L*) were exported to a computer using SPECTRA MAGIC software and the color difference (*∆E**) was calculated. The color difference (*∆E**) is a single number that represents the ‘distance’ between two colors. Color difference calculations were performed for each sample in relation with the first day and were evaluated taking into account the criteria indicated by [19]: For *ΔE** < 1 the color differences are not appreciated by the human eye, 1 < *ΔE** < 3 the human eye can see minimal color differences depending on the angle, *ΔE** > 3 the color differences are obvious to the human eye.

#### 2.5.3. Syneresis Determination

Syneresis was considered an indicator of water holding capacity. The methodology of [20] was applied. Briefly, the Eppendorf tubes containing the gels were weighed. Then, they were centrifuged in a micro-centrifuge (Eppendorf 5417CR, Eppendorf AG Hamburg Germany)). The conditions were 1500 g for 15 min at 4 °C. The released liquid was drained and the sample was weighed again to determine by difference in weight and thus sample liquid loss. The eppendorf tube was then weighed alone to obtain the initial gel weight according to the difference. Syneresis was expressed as a percentage of total water. Determinations were made in triplicate.

#### 2.5.4. Determination of Firmness

A TA-XTplus texturometer (Stable Micro Systems, Godalming, UK) was used. A compression test was performed with the P/75 probe, at a speed of 1 mm/s and a deformation of 30%. Samples were removed from eppendorf tubes where jellification had taken place and cylindrical probes of 0.5 cm diameter and 0.5 cm height were prepared for each test. Sets of three cylindrical probes were used for each condition tested. Firmness was considered as the maximum peak force that occurs during compression. TEXTURE EXPONENT 3 Version 6.1.18.0 software was used for data analysis.

### 2.6. Statistical Analysis

The data reported are an average of triplicate experiments for each sample. Statistical calculations: mean and standard deviation, as well as ANOVA were carried out by using Statgraphics Centurion XVII.II (Statpoint Technologies Inc., Warrenton, VA, USA).

## 3. Results

### 3.1. Effect of Extraction on the Protein Content

Table 1 shows the protein content in the different fractions (residual, pellet and supernatant) after acid at pH 5 and alkaline at pH 10 extractions from cricket flour. Significant differences (*p* ≤ 0.05) were observed depending on the extraction method and the extraction fraction (Table 1). For both residuals, acid and alkaline extraction, the protein content was similar, no significant differences were observed (*p* > 0.05). The highest protein content was found in pellets when the extraction was carried out under alkaline conditions (71.60% dry weight (dw)), (*p* ≤ 0.05). The lowest protein content was found in the supernatant after alkaline extraction (55.13% dw) (*p* ≤ 0.05).

### 3.2. Study of Gelation Induced by Enzymes, Heat and High Hydrostatic Pressure

The gelling properties of the proteins extracted in the different fractions and conditions were investigated. Results from visual observation of gelation of the different fractions obtained are shown in Table 2. In general, the pellet samples from the two extraction types tend to form a pasty structure, which splits into two phases after 48 h of storage. The supernatant fraction (soluble proteins) increased its viscosity without gelling, even when heat and high hydrostatic pressure (HHP) were applied. Regarding the residual fraction, no increase in, pastiness or viscosity, or gelation was observed. As expected, high hydrostatic pressure produced gelation of the whey protein isolate (WPI) used as control.

### 3.3. Gelation Induced by Gelatin and High Hydrostatic Pressure

After the previous studies which were unable to obtain stable gels, with only slight increases in viscosity, we considered the incorporation of protein extracts in gelatin as an alternative to obtaining gels for new foods. Gels were formulated using the pellet (P) and supernatant (S) fractions from each extraction type (acid-EA and alkaline-EB) and the effect of protein concentration on some physicochemical properties of the insect protein/gelatin gels obtained were studied during 28 days of storage (Appendix A).

#### 3.3.1. pH Evolution with Storage

Figure 1 shows the evolution of pH for the control sample (WPI) at different concentrations. No noticeable variations were found in pH during the storage period at any concentration level used in the study, with non-significant differences (*p* ≤ 0.05) between the pH at the beginning and the end of the storage period. Only on day 14 and 21 were the values at 11% concentration significantly different (*p* ≤ 0.05).

Evolution in pH in samples containing insect protein can be also seen in Figure 1. No noticeable differences in pH were observed throughout the storage period between samples containing insect protein belonging to the soluble fraction (EA-S, EB-S) or to the pellet (EA-P or EB-P). Non-significant differences (*p* ≤ 0.05) between the pH at the first day and the end of the storage period (day 28) were observed. However, the final pH of the gel formed depended on the type of extraction used to obtain the protein isolate lyophilized powder, so it was reflected on the pH of rehydrated samples and of gelatine-protein gels. The acid-extracted protein gels (EA) had an acidic pH while the alkaline-extracted protein gels (EB) had a basic pH value. The fact that the pH, in general, remains stable during the storage time indicates that the gels obtained by adding gelatin to the protein isolate remain stable over time, possibly indicating they do not undergo any chemical processes leading to changes in pH. In general, the pH evolution along the storage time was similar to that obtained with the whey protein isolate, used as a reference in this work.

#### 3.3.2. Instrumental Color Results

Color is one of the most important parameters when it comes to defining the stability of a new food, and its variations can influence consumer preferences. Consequently, it is an essential measure in new product development. Luminosity (L*), red/green coordinates (a*) and yellow/blue coordinates (b*) and *ΔE** as a measure assessing whether the changes in color produced were appreciable to the naked eye are shown in Table 3. L* values of the WPI gels were higher than the cricket protein gels, regardless of the extraction method used, acid or alkaline, or the fraction considered, supernatant or pellet. This was probably due to the initial color of the protein isolate, white for WPI and light grey for insect protein. These measurements correlated well with the L* parameter (luminosity), L* = 0 is black and L* = 100 is white. WPI gels had values between 72.40 and 75.78 depending on the percentage of protein for the first day of storage, while gels with insect protein had values between 21.36 and 36.72 for the first day of storage; variations on values depended on the percentage of protein and extraction method as can be seen in Table 3. In all gel samples there was a decrease in luminosity from day 1 to day 28 of storage unless this loss of luminosity was greater in the case of gel samples containing WPI than in those containing insect protein isolate. If we focus on the gels with the highest concentration of protein isolate (15%), in the gels with WPI the luminosity decreased from 75.78 to 44.63. However, in the gels with insect protein isolate, the luminosity decreased from 21.34–36.72 for the first day of storage to 13.45–21.83 for day 28, depending on the extraction method and the fraction used. These results indicate that the impact of storage on lightness was lower for gels containing insect protein than for those containing WPI. For gels with insect protein, no trend was apparent in luminosity changes related to acid or alkaline extraction or to the use of supernatant or pellet. Nor was there a trend in luminosity changes depending on the protein concentration.

The position between red and green (a*negative values indicate green while positive values indicate red), for the gel with WPI varied between −2.73 and −1.43 depending on the WPI concentration; and its position between yellow and blue (b*negative values indicate blue and positive values indicate yellow), varied between −1.81 and 4.24, depending on the WPI concentration. With respect to gels containing insect protein, all the values of a* and b* were positive, indicating that they tended towards red and yellow. Regarding the effect of storage in the color of the insect gels, in all cases the values of a* and b* decreased, but in no case they were negative.

#### 3.3.3. Gel Firmness Results

The values for gel firmness obtained with acid- and alkaline-extracted protein at different concentrations are plotted in Figure 2A (gels obtained from the soluble fraction) and Figure 2B (gels obtained from the pellet fraction), in comparison with WPI.

In all the gels, from insect protein or WPI, increased firmness was observed with storage time in comparison to the first day of sampling. In Figure 2A, the gels whose firmness increased the most by day 28 of sampling corresponded to the WPI control at 15 and 11% of protein concentration, followed by gels containing insect protein from alkaline extraction at 15%. It is noteworthy that firmness increased most in the gels containing WPI compared with gels containing insect protein from the soluble fraction. Figure 2B shows that the greatest increase in firmness was obtained in gels with insect protein at 15% from pellet produced by acid extraction.

#### 3.3.4. Syneresis Evolution during Storage

The exudate from the gels or syneresis is an indicator of the water holding capacity. A more stable gel will have higher water holding capacity and lower syneresis. Figure 3A,B shows the percentage of syneresis during gel storage. In general, the gels formed with gelatin and insect protein from the insoluble fraction (pellet) (Figure 3B) had a higher water retention capacity (lower syneresis) than gels formed with gelatin and the soluble fraction (Figure 3A), regardless of the protein-extraction method (acid or alkaline). WPI gels presented lower syneresis values than gels made with insect protein from the soluble fraction (Figure 3A).

Regarding the percentage of protein in the gels, WPI samples formulated with 11% and 15% of whey protein isolated presented lower syneresis during the storage period, although syneresis increased on day 14, mostly for the 7% protein gels, decreasing again on day 28.

In the case of gels with insect protein from soluble fraction obtained by acid extraction, formulations with 11 and 15% of protein were those with lower syneresis values, whereas when the protein was coming from alkaline extraction, the syneresis values were quite similar among the different concentrations. Similarly to WPI samples, syneresis increased on day 14, decreasing again on day 28. It is noticeable that for all protein concentrations, the gels from soluble fractions obtained by acid extraction had higher syneresis values (Figure 3A), indicating that the most suitable gels are those obtained from alkaline extraction which would respond better to the storage conditions.

For gels formulated with protein from the insoluble fraction (pellet) (Figure 3B), the results differed slightly from those obtained with the soluble fraction. In gels with protein from acid extraction, the lowest syneresis value was achieved in formulations with 15% protein, followed by gels with 7% protein. In the alkaline-extracted protein gels, the lowest syneresis values were observed in gels formulated with 11% and 15% protein, with those containing 15% protein showing the lowest syneresis on any day of storage. As in the case of WPI and the protein gels from the soluble fraction, a slight increase in syneresis was also observed on day 14 of storage.

## 4. Discussion

In general, regardless of the extraction method, protein content values were close to the values reported by [21], 65.95% in the male cricket and 65.11% in the female cricket flour. Focus on the acid extraction, results indicated that the protein content from *A. domesticus* extract fractions were comparable to those obtained by [22], being 66.66 and 65.79% dry weight for the pellet and residue fractions, respectively, applying the same method described by [14] for acid extraction. According to the mentioned works, the results obtained in this study with the methodology used for extraction are comparable to those described in the scientific literature.

No gelation was observed in cricket protein isolated in the conditions of this study. Nevertheless, different researchers have induced gelation in muscle proteins by applying transglutaminase (TGase) and glucose oxidase (GOx) as an adjuvant. It has been obtained gels treating chicken myofibrillar protein suspensions with TGase at a concentration of 0.5% after 4 h of incubation and subsequent heating at 85 °C [23]. The induction of gelation in suspensions of pig myofibrillar proteins pre-treated with 0.64% of GOx by subjecting them to 12 h of incubation and treatment with TGase at 0.4% concentration and a further 2-h incubation period has been studied by [24]. Other authors combined enzyme treatment using 0.3% TGase with HHP, 500 MPa for 30 min at 40 °C, to treat chicken breast, egg yolk and egg white [25]. Their results suggest that TGase induced cross-links between chicken myofibrillar and egg globular proteins.

The abovementioned works indicate that myofibrillar proteins had good aptitude for gel formation, and this aptitude is better if it is pre-treated with enzymes such as TGase. Insect muscles (striated and smooth) are reported to contain actin, myosin and collagen, which have a primary structure similar to that of a vertebrate protein equivalent [26]. That is supported considering the research of [27] which identified by LC/MS-MS the presence of 18 different muscular proteins in Tenebrio molitor (another edible insect) including: α-actinin-4 (106.8 kDa), myosin heavy chain (262 kDa), myosin-2 essential light chain (16.8 kDa), Tropomyosin 2 (32.5 kDa), among others. However, there could be techno-functional differences between vertebrate and insect myosin since the results of the present work indicate that true gels were not obtained under the conditions of the study, as occurs with the whey protein isolate (WPI) used as a control. A pasty structure was observed instead by using the pellet obtained in our study that could be the result of the action of enzymes and HHP on myosin.

In both WPI and cricket protein a *ΔE** > 3 was obtained after 7 days of storage, which means that the change in color would be appreciate by the naked human eye. The effect of storage time in color was different between WPI and cricket protein. Values of *ΔE** varied in increasing order for pellet fractions of insect protein (*ΔE** between 6 and 12), soluble fractions of insect protein (*ΔE** between 15 and 20) and WPI which showed the highest effect (*ΔE** between 30 and 35). On the other hand, color tends to be more stable (less pronounced *ΔE** changes during storage time) in the WPI-based gels compared to those formulated with insect protein. Variations in color during storage could be attributed mainly to oxidation reactions. Changes in the peroxide, iodine and p-anisidine indices in sun-dehydrated crickets and cooked crickets were observed by [10]. However, the extraction method applied in the present work removed most of the fatty component from the extracts; nevertheless, the cricket’s pigments may also have been subjected to this type of reaction and therefore their configuration changed.

Hardness is an important physicochemical property for gels that influence the consumer acceptability of foods, mailings in the case of new foodstuffs. Comparing the soluble fraction with the pellet fraction in the present study, results showed that in general, the gels obtained with the soluble fractions are weaker (lower firmness) than those formed with protein from the pellet, both for acid and alkaline extracted proteins. This could reflect differences on the type of protein of the supernatant and the pellet with a greater capacity to form crosslinks and retain water molecules of the proteins present in the insoluble fraction (pellet). Possibly proteins from the pellet are more abundant in sulfhydryl groups and polar amino acids. Besides amine-reactive compounds, those having chemical groups that form bonds with sulfhydryls (–SH) are the most common crosslinkers and modification reagents for protein functional properties [28]. Considering other protein rich foods, for comparison only, it is well know that the firmness of those foods varies according to their formulation. For example, burgers enriched with dietary fiber present firmness values around 4660 g and undergo an increase of 100 g during storage for 5 weeks [29], on the other hand, beef burgers with partial replacement with eggplant protein presented values of 700 to 1000 g [30]. Sausage formulated with red tilapia shows higher values than the ones mentioned above, in the order of 30,000 g, which increase (up to 50,000 g) and decrease during storage [31]; those examples could justify the trend of firmness changes observed in the present study using insect protein or WPI.

The differences between insect protein and WPI gels could be also attributed to the fact that the behavior of insect proteins, both those from the soluble fraction and those from the pellet, composed mainly of actin and myosin, differ from α or β- globulin, which represent the majority in WPI.

Syneresis, which is the spontaneous separation of liquid resulting from the contraction of the gel without the participation of external forces, is a potential phenomenon that becomes visible on the surface or edges of the gel, which has negative implications in foodstuffs in terms of consumer acceptability. The low level of syneresis (around 2%) found in the gels with 15% protein from the insoluble fraction (pellet) reflect the suitability of the incorporation of the insect protein in foods in a gel matrix.

The syneresis values obtained during storage are within the values found in different protein-based gel foods. It has been found that a 3% water release during the storage of cooked ham formulated with alginate and carrageenan [32]. It has also been found that frankfurter sausages lost up to 5.5% water in a 24-day period of refrigerated storage [33]. This would suggest that cricket protein-based gels made with the insoluble fraction and alkaline extraction, at 11 to 15%, are very suitable in terms of their lower syneresis.

## 5. Conclusions

Isolates with high protein content can be obtained based on *A. Domesticus* flour being the best alkaline route in performance and yield. The protein extracted from *Acheta domesticus* can be applied as a texture modifier in food. The texture obtained is dependent on the type of treatment. The treatment of those proteins with TGase and GOx enzymes only increases the viscosity of solutions of soluble fractions, but does not form true gels. It is technically feasible to obtain gels by adding porcine gelatin to solutions of isolated insect protein fractions and treating them with high hydrostatic pressure. Extraction fraction and concentration have an effect in gel hardness and syneresis. The highest hardness and lowest syneresis was found with the gels obtained from the insoluble fraction at 11 and 15% concentration. Considering the results obtained in this work, new food matrices of interest to the food industry can be generated based on isolated insect protein, which could be used to replace meat or fat in a wide variety of foods. Further studies on sensory properties and microbiological food safety should be carried out.

## Figures and Tables

**Figure 1 foods-10-00858-f001:**
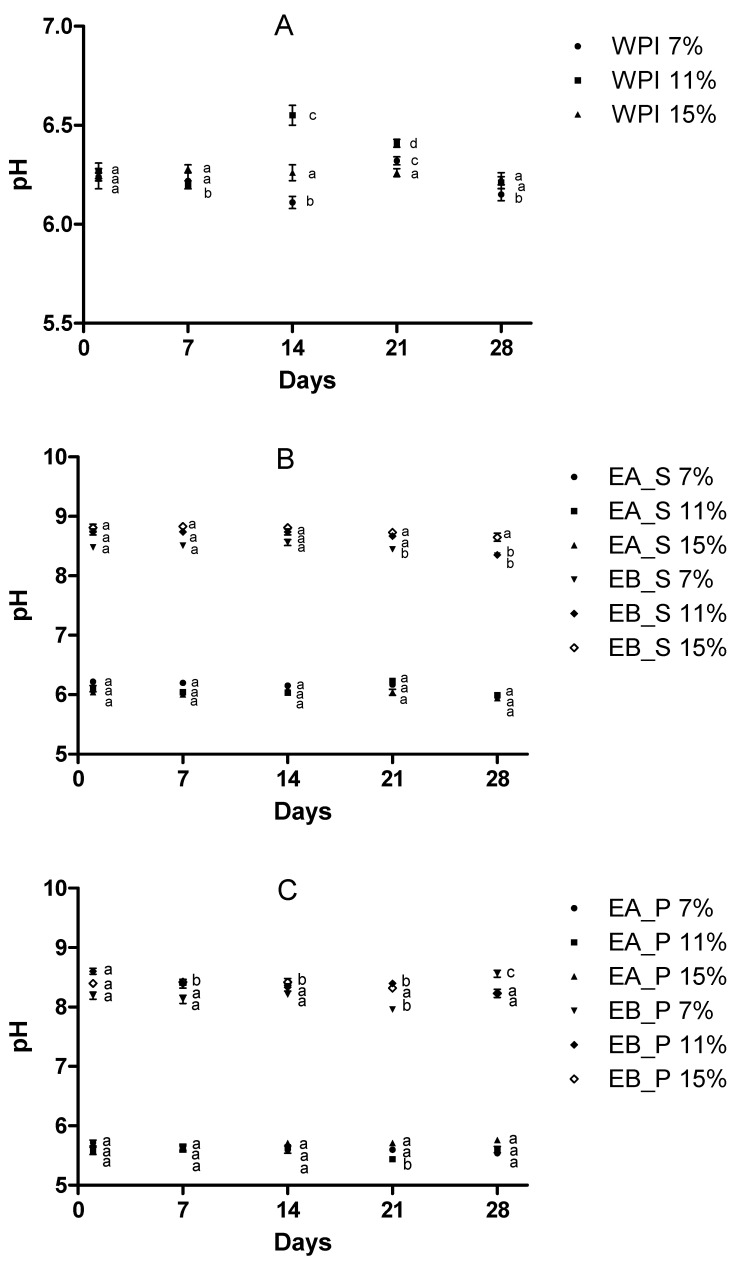
Effect of storage time on pH for WPI, soluble fraction and pellet from acid and alkaline extracted samples. Same letter means no significant differences for each substrate by days. (**A**) WPI = Whey protein isolate; (**B**) EA_S and EB_S = Soluble fraction from acid (EA) and alkaline (EB) extractions; (**C**) EA_P and EB_P = Pellet from acid (EA) and alkaline (EB) extractions. (Appendix A).

**Figure 2 foods-10-00858-f002:**
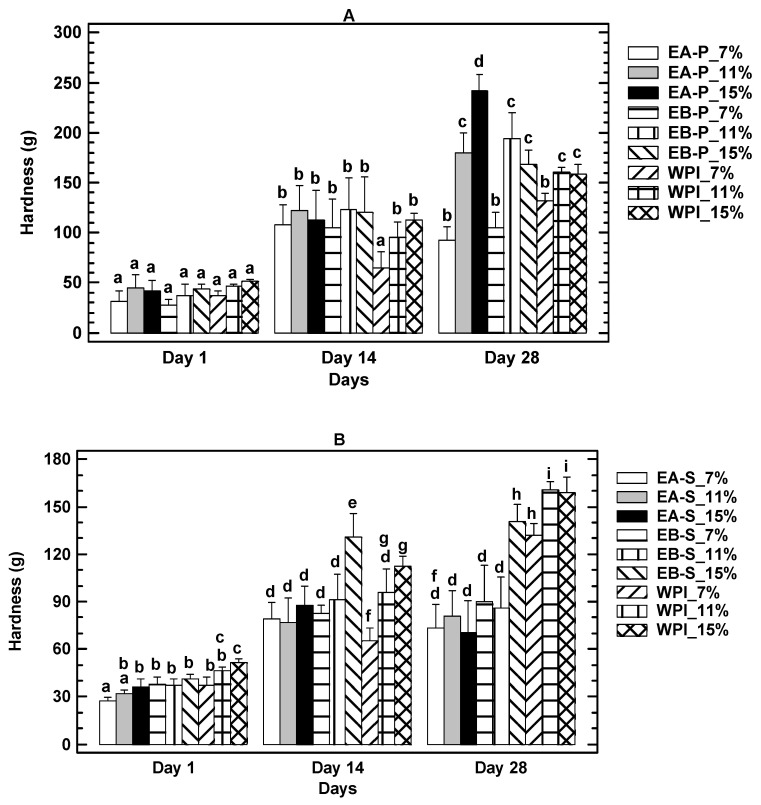
Firmness after 1, 14 and 28 days of storage (**A**) Soluble fraction, (**B**) Pellet. EA = Acid Extraction, EB = Alkaline Extraction, WPI = Whey Protein Isolate. Same letter means no significantly differences by figure. (Appendix A).

**Figure 3 foods-10-00858-f003:**
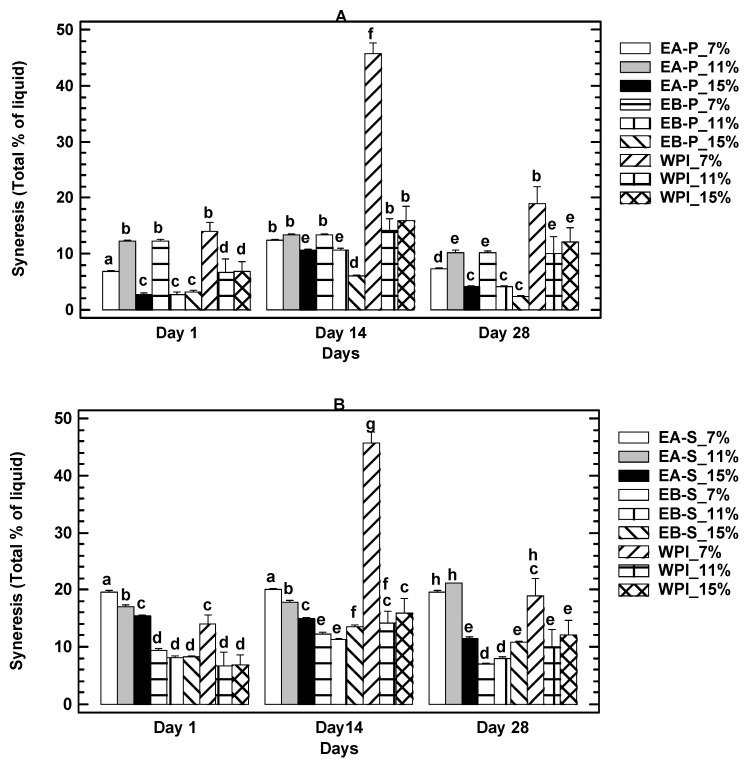
Syneresis after 1, 14 and 28 days of storage. (**A**) Soluble fraction (**B**) Pellet. EA = Acid Extraction, EB = Alkaline Extraction, WPI = Whey Protein Isolate. Same letter means no significant differences by figure. (Appendix A).

**Table 1 foods-10-00858-t001:** Percentage of cricket protein content in the different fractions obtained by acid and alkaline type extractions.

Extraction Type	Fraction	Nitrogen (% Dry Weight)	Protein (% Dry Weight)	
Acid	Residual	10.86 ± 0.08	67.84 ± 0.50	A
	Pellet	10.64 ± 0.15	66.51 ± 0.94	A
	Supernatant	9.82 ± 0.11	61.40 ± 0.69	B
Alkaline	Residual	10.69 ± 0.10	66.81 ± 0.63	CA
	Pellet	11.46 ± 0.04	71.60 ± 0.25	D
	Supernatant	8.82 ± 0.01	55.13 ± 0.06	E

Figures with same letter means that there are not significant differences (*p* ≤ 0.05).

**Table 2 foods-10-00858-t002:** Visual evaluation results of cricket protein fractions and whey protein isolate (WPI) treated with enzymes, heat and HHP.

Fraction	Acid Extraction		Alkaline Extraction		WPI	
	TGasa	GOx	TGasa	GOx	TGasa	GOx
Pellet	P	P	P	P	G	G
Supernatant	V	V	V	V		
Residual	NG	NG	NG	NG		

P: Pasty, V: Viscous, G: Gellified, NG: Non Gellified. (Appendix A).

**Table 3 foods-10-00858-t003:** Color evolution during storage on gels made by using gelatin and different isolated proteins.

Day	Substrate	Protein %	L*	a*	b*	*ΔE**
1	WPI	7	72.40 ± 0.42	−2.73 ± 0.10	−1.81 ± 0.21	0.0
		11	73.32 ± 0.62	−1.77 ± 0.21	2.32 ± 0.26	0.0
		15	75.78 ± 0.60	−1.43 ± 0.06	4.24 ± 0.38	0.0
	EA-S	7	23.27 ± 0.73	3.32 ± 0.14	13.88 ± 0.41	0.0
		11	24.36 ± 0.64	4.00 ± 0.17	15.43 ± 0.44	0.0
		15	32.84 ± 2.02	4.83 ± 0.75	17.42 ± 1.64	0.0
	EA-P	7	23.08 ± 1.10	3.27 ± 0.18	7.69 ± 0.26	0.0
		11	22.96 ± 0.81	3.54 ± 0.09	8.24 ± 0.17	0.0
		15	24.08 ± 0.91	3.43 ± 0.14	8.17 ± 0.47	0.0
	EB-S	7	33.77 ± 0.47	4.75 ± 0.14	16.81 ± 0.45	0.0
		11	30.39 ± 0.62	6.27 ± 0.21	17.76 ± 0.42	0.0
		15	36.72 ± 0.75	7.69 ± 0.21	19.00 ± 0.48	0.0
	EB-P	7	21.36 ± 0.36	3.09 ± 0.05	7.46 ± 0.09	0.0
		11	28.60 ± 1.34	2.27 ± 0.21	8.24 ± 0.54	0.0
		15	21.34 ± 0.34	2.68 ± 0.13	6.22 ± 0.29	0.0
14	WPI	7	38.86 ± 2.50	−1.06 ± 0.07	−1.31 ± 0.47	33.58
		11	38.75 ± 3.86	−0.86 ± 0.08	1.01 ± 0.59	34.61
		15	43.97 ± 1.23	−0.88 ± 0.09	1.11 ± 0.31	31.96
	EA-S	7	12.57 ± 0.73	1.61 ± 0.24	7.98 ± 0.41	12.33
		11	13.36 ± 1.19	0.64 ± 0.32	4.85 ± 0.50	15.63
		15	12.47 ± 0.54	1.63 ± 0.14	7.70 ± 0.46	22.80
	EA-P	7	14.22 ± 1.69	0.64 ± 0.68	1.80 ± 1.48	10.96
		11	14.16 ± 0.88	1.45 ± 0.50	3.64 ± 1.59	10.14
		15	18.41 ± 0.47	2.27 ± 0.11	4.96 ± 0.41	6.62
	EB-S	7	16.91 ± 1.02	0.87 ± 0.18	6.80 ± 0.33	19.99
		11	18.12 ± 1.73	1.66 ± 0.16	7.17 ± 0.82	16.86
		15	19.89 ± 1.59	3.22 ± 0.34	10.13 ± 0.33	19.54
	EB-P	7	17.11 ± 0.56	1.60 ± 0.20	4.33 ± 0.35	5.48
		11	15.23 ± 0.12	1.46 ± 0.10	4.50 ± 0.11	13.91
		15	17.46 ± 0.28	1.75 ± 0.21	4.35 ± 0.18	4.40
28	WPI	7	40.61 ± 1.13	−0.95 ± 0.04	−0.58 ± 0.91	31.86
		11	43.07 ± 2.51	−0.86 ± 0.04	0.32 ± 0.59	30.33
		15	44.63 ± 0.23	−0.85 ± 0.01	1.46 ± 0.11	31.28
	EA-S	7	10.03 ± 0.23	0.07 ± 0.10	2.70 ± 0.29	17.63
		11	11.59 ± 0.32	0.41 ± 0.08	3.81 ± 0.13	17.64
		15	13.45 ± 0.34	1.44 ± 0.15	5.98 ± 0.18	22.77
	EA-P	7	13.34 ± 0.61	0.61 ± 0.31	1.66 ± 0.96	11.76
		11	12.71 ± 0.64	1.22 ± 0.14	3.35 ± 0.46	11.58
		15	15.63 ± 0.44	1.97 ± 0.13	4.53 ± 0.20	9.31
	EB-S	7	18.17 ± 0.04	0.69 ± 0.01	5.99 ± 0.07	19.42
		11	18.86 ± 0.27	1.82 ± 0.11	7.36 ± 0.39	16.16
		15	21.83 ± 0.26	2.77 ± 0.12	8.89 ± 0.24	18.66
	EB-P	7	14.33 ± 0.34	1.62 ± 0.06	4.18 ± 0.19	7.90
		11	15.32 ± 0.12	1.52 ± 0.09	4.05 ± 0.29	13.95
		15	14.41 ± 0.16	1.90 ± 0.03	4.77 ± 0.11	7.13

## Data Availability

Data is contained within the article. The information used to produce results in this study is also available as Appendix A.

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
