# Peer review of "Effect of HHP, Enzymes and Gelatin on Physicochemical Factors of Gels Made by Using Protein Isolated from Common Cricket (Acheta domesticus)"

_foods, 2021, doi:10.3390/foods10040858_

Round 1

Reviewer 1 Report

The presented manuscript entitled “Effect of HHP, enzymes and gelatin on physicochemical factors of gels made using protein isolated from Common Cricket (Acheta domesticus) concerns the application of insect proteins in food production. The subject seems very interesting and actual since insect proteins are widely studied. Unfortunately, the obtained results did not show that the isolated samples possess good techno-functional properties. Additionally, there is much confusion about the methods described, requiring clarification before further processing the manuscript.

Detailed comments:

Title: is appropriate and reflects the manuscript content.

Abstract:

L11: Please full name when the enzymes are used for the first time in the abstract. These comments also concern the main manuscript text.

L19: What type of % the authors refer to? Is it w/w?

L19: Colour should be referred to as an objective scale and numerical value.

Keywords: There are four keywords. Thus, the authors can add additional ones to increase the manuscript visibility.

Introduction: Generally, the introduction provides the necessary information, and the aim is clearly defined.

L38-39: The sentence needs rephrasing since it is not very clear.

L40-46: In this part, there is no reference – please provide an appropriate reference.

Materials and methods:

There is no information about the materials and chemicals used in the study – please add a paragraph in which all the chemicals used and their manufactures will be provided.

L95: Very generally, please provide a more detailed description and information on what was done in the preliminary analyses and what was the main criteria used in the preliminary analyses.

L100: Magnetic stirrer is not used for homogenization! It is a fundamental aspect for the sample preparation and requires clarification – such a mistake can strongly influence the results.

L122: What was the activity of the enzymes used? Whether the enzymatic reaction was stopped after incubation or not?

L139: Use g instead of grams.

L179-182: Although the statistic is described, it was not used in the manuscript (results without ANOVA analysis).

Results and discussion: The authors chased to separate the results description from the discussion.

L185-192: Only one sample was described.

L193-195: What is the purpose of this description?

Table 1: Figures? There are no letters showing statistical differences.

L224: ‘a little higher’ does not sound scientific.

L229-232: What can affect such observations? Is there any explanation? Maybe sample preparation procedure or enzymatic reaction?

Figure 1: It is not very accessible due to small pictograms.

Figure 2: Without ANOVA.

L283-286: It is a discussion rather than the description of a result – it should be moved to the next section.

L350: WPI was used as a control - the description shows that insect protein contains proteins that are similar to animal proteins (muscle proteins). Please justify the selection of such a protein for comparison.

L353-358 and L359-366: It is the description of a result, not a discussion and should be moved to another part of the manuscript.

L377-378: The authors did not use any specific procedure of lipide removal from the samples. Please clarify.

Conclusions:

L417-418: The authors did not determine exactly that enzymes can induce reversible aggregation.

Author Response

Reviewer  1

Abstract:

L11: Please full name when the enzymes are used for the first time in the abstract. These comments also concern the main manuscript text.

Added: transglutaminase (TGasa) or glucose oxidase (GOx)

L19: What type of % the authors refer to? Is it w/w?

Added: (w/v)

L19: Colour should be referred to as an objective scale and numerical value.

Sentence was changed to: Color difference (∆E). ∆E is a numerical value and objective

Keywords: There are four keywords. Thus, the authors can add additional ones to increase the manuscript visibility.

Added edible insects

Introduction: Generally, the introduction provides the necessary information, and the aim is clearly defined.

L38-39: The sentence needs rephrasing since it is not very clear.

The sentence was rephrased.

L40-46: In this part, there is no reference – please provide an appropriate reference.

This reference was added:  Cuauhtemoc Marin , Diana Ibanez , Gabriela Rios-Corripio, Jose Angel Guerrero, Dolores Rodrigo , Antonio Martinez.  Nature of the inactivation by high hydrostatic pressure of natural contaminating microorganisms and inoculated Salmonella Typhimurium and E. coli O157:H7 on insect protein-based gel particles. LWT - Food Sci Technol. 2020 https://doi.org/10.1016/j.lwt.2020.109948

Materials and methods:

There is no information about the materials and chemicals used in the study – please add a paragraph in which all the chemicals used and their manufactures will be provided.

Added as raw material and chemicals providers:

  • Cricket flour (Thailand Unique Brand, Thailand) was used for this purpose
  • (Scharlab Chemie S. A., Barcelona, Spain)

L95: Very generally, please provide a more detailed description and information on what was done in the preliminary analyses and what was the main criteria used in the preliminary analyses.

Added: pH, centrifugation speed, time and temperature as parameters for optimization

Added: to achieve the highest protein content in the pellet as main criterium

L100: Magnetic stirrer is not used for homogenization! It is a fundamental aspect for the sample preparation and requires clarification – such a mistake can strongly influence the results.

You are alright “homogenization” is no the correct wold it was substituted by “stirred to achieve a good mixture before filtration”

L122: What was the activity of the enzymes used? Whether the enzymatic reaction was stopped after incubation or not?

Transglutaminase (TGase): Used in meats to catalyse acyl group transfer reactions and induction of covalent cross-links in proteins.

Glucose oxidase (GOx): Catalyzes the oxidation of glucose to glucuronic acid using oxygen as an electron acceptor, forming a hydrogen peroxide molecule. In protein systems, the peroxide resulting from the treatment with GOx acts as an inducer of disulphide bond formation.

All samples were incubated at 50ºC for 16 hours temperature for enzymatic activity. Then, they were kept refrigerated at 4 ° C for 2 days.

L139: Use g instead of grams.

Change was done

L179-182: Although the statistic is described, it was not used in the manuscript (results without ANOVA analysis).

Anovas were carried out but no letters were present in figures.

It was corrected and letters were added to figures improving the interpretation of results.

Results and discussion: The authors chased to separate the results description from the discussion.

L185-192: Only one sample was described.

For acid or alkaline extraction three samples were described (residual, supernatant and pellet) with repetitions as described in material and methods

L193-195: What is the purpose of this description?

Sorry, that paragraph is part of the instructions to prepare the manuscript, it was deleted

Table 1: Figures? There are no letters showing statistical differences.

Letters were added to figures

Table 1 has letters, but were placed at the end of each line by the editorial officer

L224: ‘a little higher’ does not sound scientific.

The sentence was changed to be more scientific

L229-232: What can affect such observations? Is there any explanation? Maybe sample preparation procedure or enzymatic reaction?

The type of extraction acid or alkaline affected to the pH of the powder obtained after the lyophilization, so it was reflected on the pH of rehydrated samples and of gelatine-protein gels

Figure 1: It is not very accessible due to small pictograms.

Bigger pictograms mask the error bars and overlapping each other, sorry if it is a little bit difficult to read.

Figure 2: Without ANOVA.

Annovas were carried out but no letters included. It was Solved

L283-286: It is a discussion rather than the description of a result – it should be moved to the nnnext sectionnext section.

L283-286: Placed in results

L350: WPI was used as a control - the description shows that insect protein contains proteins that are similar to animal proteins (muscle proteins). Please justify the selection of such a protein for comparison.

The justification is that WPI produces very good gels, as indicated in the scientific literature, in the presence of high pressure and enzymes, so it was used as a reference to be clear that the conditions used in the present work, at least, produced the gelling of proteins, even if they were lacteal.

L353-358 and L359-366: It is the description of a result, not a discussion and should be moved to another part of the manuscript.

L353-358: Placed in Results

L359-366: Placed in Results

L377-378: The authors did not use any specific procedure of lipide removal from the samples. Please clarify.

For the two extractions, the fatty matter that remains floating in the supernatant is removed with a spoon. It was included in material and methods.

Conclusions:

L417-418: The authors did not determine exactly that enzymes can induce reversible aggregation.

Aggregation was not determined buy it is a hypothesis, we think that the behaviour could be due to aggregation, nevertheless the sentence was deleted.

Reviewer 2 Report

This research investigates the effect of high hydrostatic pressure, transglutaminase, glucose oxidase and gelatin on physicochemical factors of gels made by using common cricket isolated protein obtained by acid and alkaline extractions. It is an interesting topic for novel food processing industry. Although the use of alkaline for increasing cricket protein extraction, the sensory properties of cricket protein made from different fractions are not evaluated. Transglutaminase and glucose oxidase enzymes con induce reversible aggregation in dispersions of insoluble protein fractions and increase viscosity in dispersions of soluble fractions. These enzymes do not induced gel formation even after HHP treatment. The amino acid profile of cricket protein isolates should be investigated to explain why enzymatic treatment on cricket protein can only increase viscosity but not to form gels comparing to whey protein isolate. The introduction and materials and methods were well written, nevertheless, there was a few mistakes should be corrected as attached file. The PhD Thesis of school and city of reference (5) should add. The paper is clear and able to read by the focus groups.
The conclusions are well present but I suggest combining to one paragraph. Proposed experimental conditions and results could gain high correlated and it is reliable. 

Author Response

Reviewer 2

Line 11: Full names of enzymes

Done

Line 68: Remove the comma

Done

Line 123: TGase

Done

Line 132: [16]

Done

Line 142: What ® ROVAC standing for?

Coextruded ROVAC ® films for High Hydrostatic Pressure treatments. ROVAC ® films preserve the aroma, the colour and freshness of food for a longer period and have good transparency and puncture resistance.

Line 144: Remove the comma

Done

Line 188: Table 1, T upper case

Done

Line 202: Table 2, T upper case

Done

Line 238: Table 3, T upper case

Done

Line 246: Table 3, T upper case

Done

Line 277: Figure 2A, F upper case

Done

Conclussions: Is suggested to be one paragraph

Done

Line 448: School and city for PhD thesis

Escuela Agrícola Panamericana, Zamorano,  Honduras

Line 506: Italic

Done

Round 2

Reviewer 1 Report

The manuscript was corrected following the recommendations.